# Comprehensive Analysis of the INDETERMINATE DOMAIN (IDD) Gene Family and Their Response to Abiotic Stress in *Zea mays*

**DOI:** 10.3390/ijms24076185

**Published:** 2023-03-24

**Authors:** Xue Feng, Qian Yu, Jianbin Zeng, Xiaoyan He, Wujun Ma, Lei Ge, Wenxing Liu

**Affiliations:** 1The Characteristic Laboratory of Crop Germplasm Innovation and Application, Provincial Department of Education, College of Agronomy, Qingdao Agricultural University, Qingdao 266109, China; 2Department of Agronomy, College of Agriculture and Biotechnology, Zijingang Campus, Zhejiang University, Hangzhou 310058, China; 3State Agricultural Biotechnology Centre, College of Science, Health, Engineering and Education, Murdoch University, Perth, WA 6150, Australia; 4The Key Laboratory of the Plant Development and Environmental Adaptation Biology, School of Life Sciences, Ministry of Education, Shandong University, Qingdao 266237, China

**Keywords:** maize (*Zea mays*), IDD, genome-wide, expression patterns, regulatory network

## Abstract

Transcription factors (TFs) are important regulators of numerous gene expressions due to their ability to recognize and combine cis-elements in the promoters of target genes. The INDETERMINATE DOMAIN (IDD) gene family belongs to a subfamily of C2H2 zinc finger proteins and has been identified only in terrestrial plants. Nevertheless, little study has been reported concerning the genome-wide analysis of the *IDD* gene family in maize. In total, 22 *ZmIDD* genes were identified, which can be distributed on 8 chromosomes in maize. On the basis of evolutionary relationships and conserved motif analysis, ZmIDDs were categorized into three clades (1, 2, and 3), each owning 4, 6, and 12 genes, respectively. We analyzed the characteristics of gene structure and found that 3 of the 22 *ZmIDD* genes do not contain an intron. Cis-element analysis of the *ZmIDD* promoter showed that most *ZmIDD* genes possessed at least one ABRE or MBS cis-element, and some *ZmIDD* genes owned the AuxRR-core, TCA-element, TC-rich repeats, and LTR cis-element. The Ka:Ks ratio of eight segmentally duplicated gene pairs demonstrated that the *ZmIDD* gene families had undergone a purifying selection. Then, the transcription levels of *ZmIDDs* were analyzed, and they showed great differences in diverse tissues as well as abiotic stresses. Furthermore, regulatory networks were constructed through the prediction of ZmIDD-targeted genes and miRNAs, which can inhibit the transcription of *ZmIDDs*. In total, 6 *ZmIDDs* and 22 miRNAs were discovered, which can target 180 genes and depress the expression of 9 *ZmIDDs*, respectively. Taken together, the results give us valuable information for studying the function of ZmIDDs involved in plant development and climate resilience in maize.

## 1. Introduction

To survive better, terrestrial plants have evolved many mechanisms to accommodate a variety of complex environments, including drought, salt, and low temperature stress. Transcription factors (TFs) play dominant roles in plant growth and development and response to abiotic stress [1,2]. TFs can recognize and combine cis-elements, thus activating or repressing the transcription of downstream genes [3]. A number of important TF families have been identified in plants, including C2H2 (Cys2His2 zinc finger) [4], WRKY [5], bHLH (basic helix-loop-helix) [6], MYB [7], bZIP (basic region-leucine zipper) [8], and MADS-box [9]. Of these, the INDETERMINATE DOMAIN (IDD) family, a class of C2H2 zinc finger proteins, has been found only in land plants [10,11].

The *IDD* gene family is characterized by the INDETERMINATE (ID) domain, which contains four zinc fingers (ZFs) [10,12]. The four ZFs include two C2H2-type ZFs involved in DNA binding and two C2HC-type ZFs responsible for protein interaction [13]. The first *IDD* genes, ZmID1, were identified in Zea mays through transposon insertion, which regulates flowering time via a leaf-generated signal [14]. The *IDD* family gene has been discovered in numerous plants, including *Arabidopsis* [15], rice [16], *Moso bamboo* [17], and so on. So far, 16 *IDD* genes have been found in *Arabidopsis*, and 12 of them have been functionally verified [18]. In *Arabidopsis thaliana*, IDDs function in various metabolic and development processes, including root and leaf development, flowering time, seed maturation, hormone signaling, and abiotic stress [19]. For example, six *AtIDD* genes (*IDD2*, *IDD3*, *IDD4*, *IDD5*, *IDD9*, and *IDD10*) are involved in gibberellin (GA) homeostasis, thus modulating florescence through interactions with DELLA proteins [20]. Coelho et al. [18] reported that *AtIDD14* could be alternatively spliced and produce two transcripts. AtIDD14α exists in normal conditions, but AtIDD14β accumulates when plants are subjected to cold stress. Moreover, AtIDD14 can also interact with ABFs/AREBs and cooperatively mediate ABA-dependent drought tolerance [21]. This suggests that IDDs are responsible for responding to different environmental stimuli in plants, but research concerning IDDs remains rare.

IDDs also play vital roles in the grass family, such as rice and barley. Ghd10, the ortholog of ZmID1, participates in mediating yield component characters via increasing plant height and tillering under short-day (SD) conditions [22]. OsIDD2 is a negative regulator in second cell wall formation through repressing target genes, which are related to sucrose metabolism and lignin synthesis [23]. OsIDD3/ROC1 enhances cold tolerance by directly targeting cis-elements of dehydration-responsive element-binding protein (DREB)/CBF1 [24]. OsIDD10 was reported to regulate ammonium absorption and nitrogen metabolism in roots [25]. In barley, BROAD LEAF1 (BLF1) was found to regulate leaf size by restraining longitudinal cell proliferation [26].

Maize is the most common cereal crop in the world. Nevertheless, genome-wide analysis of *IDDs* has not been conducted in maize. Thus far, three IDDs have been characterized in maize. For example, ZmID1 can regulate flowering time [11], and its paralogs ZmIDD9 and ZmIDDveg9 control endosperm development [27,28]. With the rapid development of sequencing technology, whole-genome analysis of *ZmIDD* gene families becomes feasible. In this study, we identified 22 candidate *ZmIDD* genes through bioinformatics analysis. Then, a systematic analysis of the phylogenetic tree, chromosome localization, gene structures, conserved motifs, and promoter cis-elements was conducted. In addition, tissue-specific and abiotic stress-mediated expression profiles of *ZmIDDs* were analyzed. At last, miRNA-target *ZmIDDs* and ZmIDDs-target genes were predicted. The results will give us valuable information for better comprehending the function of ZmIDDs in plant development and response to environmental stress in maize.

## 2. Results

### 2.1. Identification and Evolution Analysis of ZmIDD Family Genes in Zea mays

In this study, we identified 22 *ZmIDDs*. The gene and protein characteristics, such as gene names, ID, chromosomal locations, amino acid numbers, molecular weights (MW), and isoelectric points (pI), were shown in Appendix A. For instance, the amino acid length of 22 ZmIDD proteins ranges from 354 to 815. The molecular weight ranges from 38 to 90 kDa. The coding sequence and protein sequence can be seen in Appendix A. Evolutionary relationships of *IDD* genes were shown via a phylogenetic tree using 15 rice IDDs, 16 Arabidopsis IDDs, and 22 maize IDDs using the maximum likelihood method, the minimum evolution method, and the Neighbor-Joining method (Figure 1; Appendix A; Appendix A). Three major clades were identified. Clades 1, 2, and 3 contained 4, 6, and 12 *ZmIDDs*, respectively (Figure 1; Appendix A).

### 2.2. Structure Analysis of ZmIDD

The DNA structure analysis showed that *ZmIDDs* had 1–4 exons distributed unevenly (Figure 2B; Appendix A). Meanwhile, eight conserved motifs were found in ZmIDDs and named motifs 1–8 (Appendix A). In detail, all ZmIDDs have motif 1, 2, and 3. Motifs 4, 5, and 7 were presented in all subfamilies. Motif 8 only exists in the members of clade 1. Motif 6 exists exclusively in the members of clade 2 and clade 3 (Figure 2C). Multi-sequence alignment of ZmIDDs demonstrated that all ZmIDD proteins owned two C2H2-type (ZF1 and ZF2) and two C2HC-type (ZF3 and ZF4) ZFs (Figure 3). In addition, 3D structures of ZmIDD1, ZmIDD2, and ZmIDD4 proteins, belonging to clade 3, clade 1, and clade 2, respectively, were predicted, indicating each ZmIDD had different 3D structures in spite of having four ZF fingers in all ZmIDDs (Figure 4).

### 2.3. Location and Duplication of ZmIDDs

The chromosome localization showed that 22 *ZmIDDs* were distributed across the maize genome except for chromosome 4 and 6 (Figure 5). In detail, *ZmIDD* clade 1 genes were distributed on chromosomes 1, 7, and 9; *ZmIDD* clade 2 genes were found on chromosomes 1, 2, 5, 7, and 10; and *ZmIDD* clade 3 genes were localized on chromosomes 1, 2, 3, 5, 7, and 8. Then, we detected gene duplication and found 8 segmental duplication events in *ZmIDDs* (Figure 6A; Appendix A). It indicated that segmental duplication was the primary reason for the enlargement of *ZmIDD* genes in maize. Furthermore, a synteny analysis between the maize and rice genomes was also performed. It found that three *IDD* gene pairs (*ZmIDD13/OsIDD9*, *ZmIDD1/OsIDD9*, and *ZmIDD17/OsIDD11*) were found in *Zea mays* and *Oryza sativa* (Figure 6B; Appendix A).

To illustrate the evolutionary constraints acting, we calculated the Ks value, Ka value, Ka:Ks ratio, and divergence time of paralogous *IDD* genes. The majority of Ka:Ks ratios in segmental duplicated *ZmIDD* gene pairs were less than one, with the exception of *ZmIDD15/ZmIDD3* with 1.1, and divergence time occurred between 6.788 Mya and 112.101 Mya ago (Appendix A).

### 2.4. Cis-Elements Analysis in ZmIDDs Promoters

To figure out the function and regulatory pattern of *ZmIDD* genes, we scanned the promoter sequence of 22 *ZmIDDs* to analyze cis-elements, such as ABRE, AuxRR-core, MBS, TCA-element, TC-rich repeats, and LTR, related to ABA, auxin, drought-inducibility, salicylic acid, defense and stress, and low temperature responses (Figure 7; Appendix A). Generally, 15 *ZmIDDs* (68%) had ABRE cis-elements, 7 *ZmIDD* genes (32%) owned AuxRR-core elements, 12 *ZmIDD* genes (46.7%) owned MBS cis-elements, and 3 (14%) *ZmIDD* genes carried LTR cis-elements. Six *ZmIDD* genes owned TCA elements, and four *ZmIDD* genes carried TC-rich repeats.

### 2.5. Tissue-Specific Expression Patterns of ZmIDDs

A number of reports have shown that *IDD* genes are expressed in many tissues of plants. For instance, some *IDDs* are mainly expressed in mature leaves [29], and some in immature leaves [10] or roots [30]. To determine the tissue-specific expression profiles of *ZmIDDs*, the transcription levels of 22 *ZmIDDs* in six tissues of B73, such as roots, leaves, stems, embryo, endosperm, and pericarp, were compared. Based on the difference in expression patterns, 22 *ZmIDD* genes were categorized into three groups (Figure 8A; Appendix A). Group 1 consists of one gene (*ZmIDD9*), which is not expressed in six tissues. Group 2 contains 11 genes expressed only in certain tissues. For example, *ZmIDD13* was expressed only in the root but not in other tissues. Group 3 had 10 genes expressed in all tissues. Moreover, group 3 could be divided into two subgroups. Subgroup 1 contains four *ZmIDDs* with a high transcription level (log2TPM + 1 > 1) in six tissues, including *ZmIDD8*, *ZmIDD14*, *ZmIDD17*, and *ZmIDD22*. The rest of the six genes belong to subgroup 2.

### 2.6. Stress-Induced Expression Patterns of ZmIDDs

We collected transcriptome data from *ZmIDD* genes in maizeGDB and compared gene expression patterns when plants were subjected to drought, salt, heat, cold, and ultraviolet light. Generally, the expression of *ZmIDD* genes displayed significant differences under diverse abiotic stresses (Figure 8B; Appendix A). Interestingly, most *ZmIDD* genes showed down-regulation or no change in response to abiotic stress except for *ZmIDD8* and *ZmIDD21*. For example, *ZmIDD8* and *ZmIDD21* exhibited up-regulation after salt and cold stress, respectively. Meanwhile, many *ZmIDD* genes were inhibited when plants were subjected to abiotic stress. For instance, the expression of *ZmIDD6* and *ZmIDD11* was decreased under cold and drought treatments, respectively. *ZmIDD15* showed down-regulation under heat, salt, and cold stress. However, there are some *ZmIDDs* that were not in response to all the abiotic stresses analyzed in the current study, such as *ZmIDD7*, *ZmIDD9*, *ZmIDD10*, *ZmIDD13*, and *ZmIDD19*. In addition, *ZmIDD8* displayed opposite expression patterns under heat and salt stress.

### 2.7. ZmIDDs-Regulated Genes and miRNA-Targeted ZmIDDs

There were 6 ZmIDDs (ZmIDD 1, 4, 5, 9, 10, and 13) that could bind the cis-elements and mediate the transcription of 180 downstream genes in maize. The detailed information is listed in Appendix A. The GO enrichment analysis concerning downstream genes demonstrated that five ZmIDD-targeted genes were involved in protein dimerization activity (GO:0046983) and four ZmIDD-targeted genes were directed hydrolase activities, hydrolyzing O-glycosyl compounds (GO:0004553) (Figure 9; Appendix A).

To determine whether a miRNA-mRNA regulatory network exists in maize, 321 known miRNAs were scanned. Finally, 22 miRNAs were predicted to inhibit the transcription level of *ZmIDDs* (Figure 10; Appendix A). In short, miRNA-targeted *ZmIDDs* were divided into 5 networks: group 1/4/5/6 involving *ZmIDD* 4/7/18/5, respectively; group 2 involving two *ZmIDDs* (*ZmIDD8* and *ZmIDD17*); and group 3 involving three *ZmIDDs* (*ZmIDD* 2/3/21) (Figure 10; Appendix A).

## 3. Discussion

Transcription factors have regulatory and functional roles in plant growth and development [31]. The indeterminate domain (*IDD*) genes exist universally in all plants. In *Zea mays*, only three IDDs have been functionally verified: INDETERMINATE1 (ID1), the dominant regulator in flowering [14,32], and Naked Endosperm 1 and 2 (NKD1 and NKD2) associated with seed development [26,28]. Hence, whole-genome analysis and function prediction of the *ZmIDD* genes were necessary in maize. In this study, we identified 22 *ZmIDD* genes, which were named *ZmIDD1* to *ZmIDD22* according to their chromosome localization in maize (Figure 5; Appendix A). Three *IDDs* (*ID1*, *NKD1*, and *NKD2*) correspond to *ZmIDD5*, *ZmIDD7*, and *ZmIDD22* identified in this work, respectively. Gene duplication and diversification play an essential role in plant evolution [33]. Prochetto et al. [19] reported that IDD derived from Chalorophyta and experienced a gene duplication event about 470 million years ago (MYA). Several times, duplication led to the emergence and diversification of terrestrial plants. Hence, monocots and dicots owned the same *IDD* gene numbers (16–23) [10,34,35]. In the current study, eight segmental duplication events were found in *ZmIDDs* (Figure 6; Appendix A). Meanwhile, divergence time occurred between 6.788 Mya and 112.101 Mya ago, and most Ka/Ks values were less than one (Appendix A), implying that the *ZmIDD* gene family may have suffered robust purifying selective pressure in the course of evolution.

Plant hormones, as important signaling molecules, can control plant growth, development, and stress responses [36]. Previous research has reported that many IDDs are related to hormone homeostasis [12,37,38]. For example, IDD14, IDD15, and IDD16 can cooperatively function in lateral organ morphogenesis and gravitropism through accelerating auxin synthesis and transport in *Arabidopsis* [39]. As shown in Figure 1, Appendix A, *AtIDD14* was a paralog of *AtIDD15* and *AtIDD16*, and *ZmIDD2*, *ZmIDD3*, *ZmIDD6*, *ZmIDD15*, and *ZmIDD21* were orthologous genes of *IDD14*, *IDD15*, and *IDD16* in *Arabidopsis*. Meanwhile, the promoter of 7 *ZmIDDs* has an AuxRR-element involved in auxin biosynthesis, including *ZmIDD1*, *ZmIDD2*, *ZmIDD6*, *ZmIDD7*, *ZmIDD9*, *ZmIDD13*, and *ZmIDD15* (Figure 7; Appendix A). It was suggested that although *ZmIDD3* and *ZmIDD21* were orthologs of *AtIDD14*, the promoter of *ZmIDD3* and *ZmIDD21* does not have the AuxRR-element, thus developing different functions in plant growth. Wang et al. [40] reported that miRNA167 could directly regulate the auxin response factors GmARF8a and GmARF8b and participate in lateral root development in soybean. In this study, miR167 could target and lead to the degradation of *ZmIDD7* (Figure 10; Appendix A). In addition, xyloglucan endotransglucosylase/hydrolase (XTH) is regulated by auxin and functions in plant developmental plasticity [41]. Gullner et al. [42] reported that glutathione-S-transferases (GSTs) were involved in intracellular auxin transport. Here, ZmIDD1 and ZmIDD13 could target xyloglucan endotransglucosylase/hydrolase protein 32 and glutathione S-transferase GST 21 (Appendix A). The results demonstrated that ZmIDD may also be responsible for auxin biosynthesis, thus regulating plant growth.

Except for the zinc finger domain, the IDD proteins owned two conserved amino acid residues (Ser73 and Ser182): Ser73 could be phosphorylated by MPK6 [38], and Ser182 has been reported to be modified by AKIN10 (the catalytic subunit of SnRK1) [37]. Here, all 22 ZmIDDs have Ser73 and Ser182 (Figure 3). Meanwhile, the amino acid length of 22 ZmIDD proteins ranged from 354 to 815 (Appendix A). It indicated that although the ZmIDDs contained a highly similar ID domain in their N-terminal, the flank sequences varied greatly. Jeong et al. [37] reported that phosphorylation of AtIDD8 at Ser-182 obviously decreased its transcriptional activity, and atidd8 mutants or over-expression of AKIN10 led to a delay in flowering in Arabidopsis. In the current study, protein-protein interactions (PPI) between ZmIDDs and ZmSnRKs were predicted using the STRING web server. Only one ZmSnRK protein (ZmSnRK1.1) that could interact with the ZmIDD protein (ZmIDD5) was found (Appendix A). ZmIDD5 exhibited expression in most tissues except for the root and showed down-regulation under cold and heat stress (Figure 6; Appendix A). Meanwhile, miR156 might bind and cause the cleavage of *ZmIDD5* (Figure 10; Appendix A). It was reported that miR156 could enhance cold and heat stress tolerance by repressing the expression level of transcription factors in plants [43,44]. It suggests that ZmIDD5 may also respond to abiotic stresses and regulate flowering time. The starch degradation in guard cells can cause stomatal opening [45], which is likely to increase guard cell turgor pressure by supplying soluble sugars. Furthermore, Seo et al. [30] discovered that AtIDD14 directly activated qua-quine starch, thereby mediating starch metabolism in *Arabidopsis*. Hence, IDD14-regulated starch degradation is likely to be responsible for stomatal opening. In this study, *ZmIIDD6* and *ZmIDD11* were down-regulated under drought stress, but other *ZmIDDs* do not respond to drought responses (Figure 8; Appendix A). The promoters of *ZmIDD6* and *ZmIDD11* have an ABRE element and a MBS element, respectively (Figure 7; Appendix A). It indicated that ZmIDD could also respond to drought stress in maize. The Indeterminate Domain Protein ROC1 (the ortholog of *ZmIDD21*) enhanced chilling tolerance by activating DREB1B/CBF1 in rice [24]. Phytochrome A signal transduction 1 (PAT1) could interact with IDD3 to activate lipoxygenase 3 (LOX3) and increase JA-Ile accumulation in *grape calli* under low temperature stress [46]. In the current study, *ZmIDD21* was up-regulated under cold stress (Figure 8; Appendix A), implying ZmIDD21 was also a positive regulator of cold stress.

Generally, we conducted a genome-wide analysis of *IDD* genes in maize. It has a vital implication for further comprehending the biological functions of ZmIDDs. However, a lot of functional verification work needs to be conducted in future studies.

## 4. Materials and Methods

### 4.1. Identification of ZmIDDs in Zea mays

The genome and protein sequences of maize B73 were obtained from MaizeGDB (https://maizegdb.org/ (accessed on 12 October 2022)). With default parameters, the Hidden Markov Model (HMMER3.0) profile of the C2H2 protein domain (PF00096) was used to search the protein database in the maize genome [47]. Then, we used the maize genome database to conduct BLASP. Gene and protein sequences in *Arabidopsis* were downloaded from the Ensemble Plants database [48]. The longest transcripts were retained, and incomplete sequences were deleted. In addition, we used the SMART database to reconfirm sequences [49]. Putative *ZmIDD* genes were named *ZmIDD1* to *ZmIDD22* based on their chromosome localization. Moreover, protein characteristics were determined by the ExPASy tool (http://www.expasy.org/tools/ (accessed on 20 October 2022)) [50].

### 4.2. Phylogenetic Tree, Gene Structure, Conserved Motifs, and Cis-Elements Analysis

Multiple protein sequence alignments were conducted using ClustalW [51]. A phylogenetic tree was constructed by MEGA version 7.0 through the maximum likelihood method, Neighbor-Joining method, and Minimum Evolution method [52]. Firstly, the evolutionary history was inferred by using the Maximum Likelihood method based on the Jones-Taylor-Thornton (JTT) matrix-based model. Initial tree(s) for the heuristic search were obtained automatically by applying Neighbor-Join and BioNJ algorithms to a matrix of pairwise distances estimated using a JTT model and then selecting the topology with the highest log likelihood value. The tree is drawn to scale, with branch lengths measured in the number of substitutions per site. Secondly, evolutionary history was inferred using the Minimum Evolution method. The percentage of replicate trees in which the associated taxa clustered together in the bootstrap test (1000 replicates) is shown next to the branches. The evolutionary distances were computed using the p-distance method and are in units of the number of amino acid differences per site. The ME tree was searched using the Close-Neighbor-Interchange (CNI) algorithm at a search level of 1. The tree is drawn to scale, with branch lengths in the same units as those of the evolutionary distances used to infer the phylogenetic tree. Finally, evolutionary history was inferred using the Neighbor-Joining method. The percentage of replicate trees in which the associated taxa clustered together in the bootstrap test (1000 replicates) is shown next to the branches. The evolutionary distances were computed using the p-distance method and are in units of the number of amino acid differences per site. In addition, we used iTOL to beautify the evolutionary tree (https://itol.embl.de/ (accessed on 21 October 2022)) [53].

The DNA structure, such as exon and intron arrangement, was detected by the Gene Structure Display Server (GSDS) tool (http://gsds.cbi.pku.edu.cn/ (accessed on 21 October 2022)) [54]. The conserved motifs were analyzed by the MEME program with default parameters [55] and annotated using the InterProScan database (http://www.ebi.ac.uk/Tools/pfa/iprscan/ (accessed on 1 November 2022)) [56]. The gene structures and motifs were then imaged using the TBtools software (v1.09832) [57].

Promoter sequences (−1.5 kb) of each *ZmIDD* gene were used to scan any potential cis-elements by PlantCARE (http://bioinformatics.psb.ugent.be/webtools/plantcare/html/ (accessed on 12 November 2022)) [58]. Then, we used TBtools to visualize the architecture [57].

### 4.3. Chromosome Distribution and Synteny Block of ZmIDDs

We used the Circos tool to locate *ZmIDD* genes on maize chromosome [59]. The collinearity of the orthologous *IDD* genes between maize and rice was analyzed using MCScanX software [60]. The nonsynonymous substitution rate (Ka) and synonymous substitution rate (Ks) of ZmIDDs were calculated by the ParaAT tool [61]. The Ka:Ks ratio was then calculated using Calculator 2.0 software [62]. In addition, Ks/2λ was used to estimate gene duplication time [63], where λ = 1.5 × 10 − 8.

### 4.4. Modeling of 3D Structures of ZmIDDs and Zinc Finger Alignment

We used Swiss-Model (https://swissmodel.expasy.org/interactive/ (accessed on 12 January 2023)) to predict the 3D structure of ZmIDD proteins and the SAVES server to assess the 3D structure (http://saves.mbi.ucla.edu/ (accessed on 12 January 2023)) [64]. Meanwhile, the amino acid alignment of the zinc finger was performed by DNAMAN.

### 4.5. Prediction of ZmIDDs-Regulated Genes

To better comprehend the regulatory mechanism of ZmIDDs, the target genes regulated by ZmIDDs were predicted through the online PlantRegMap tool [65], where the species and parameters were set to *Zea mays*, Organ:All, Method:FunTFBS, and Mode:TF (retrieve targets). Then, Gene Ontology (GO) enrichment of target genes was analyzed by an online bioinformatics server (http://www.bioinformatics.com.cn/?p=1 (accessed on 1 February 2023)) with a *p*-value < 0.05.

### 4.6. Prediction of miRNA-ZmIDD Regulatory Networks

The miRNAs, which can regulate the *ZmIDD* gene expression, were predicted using the psRNATarget server with the following parameters: the penalty for a G:U pair was one, and the number of mismatches allowed in the seed region was zero [66]. Then, we used Cytoscape V3.8.2 software (https://cytoscape.org/download.html (accessed on 1 February 2023)) to image the interaction networks.

### 4.7. Expression Profile Analysis of ZmIDDs

The tissue-specific and abiotic stress-induced transcriptional data of *ZmIDDs* in maize B73 were downloaded from qTeller in MaizeGDB [67,68]. We used the DSEeq2 R package to conduct differential expression analysis, and the heatmaps were imaged by TBtools software. The genes with a |log2 ratio|≥1 were considered differentially expressed genes (DEGs).

## 5. Conclusions

In the current study, we identified and characterized 22 ZmIDD proteins that contained a complete IDD domain. On the basis of their amino acid sequences, the 22 ZmIDDs were categorized into three clades. In the course of the evolution of *ZmIDD* genes, segmental duplication events played a dominant role. Moreover, the cis-acting elements, gene expression, and regulatory network of ZmIDD families were also analyzed. These findings add to our understanding of the *ZmIDD* gene family’s characteristics and provide useful information for further functional characterization of ZmIDDs in maize climate resilience. 

## Figures and Tables

**Figure 1 ijms-24-06185-f001:**
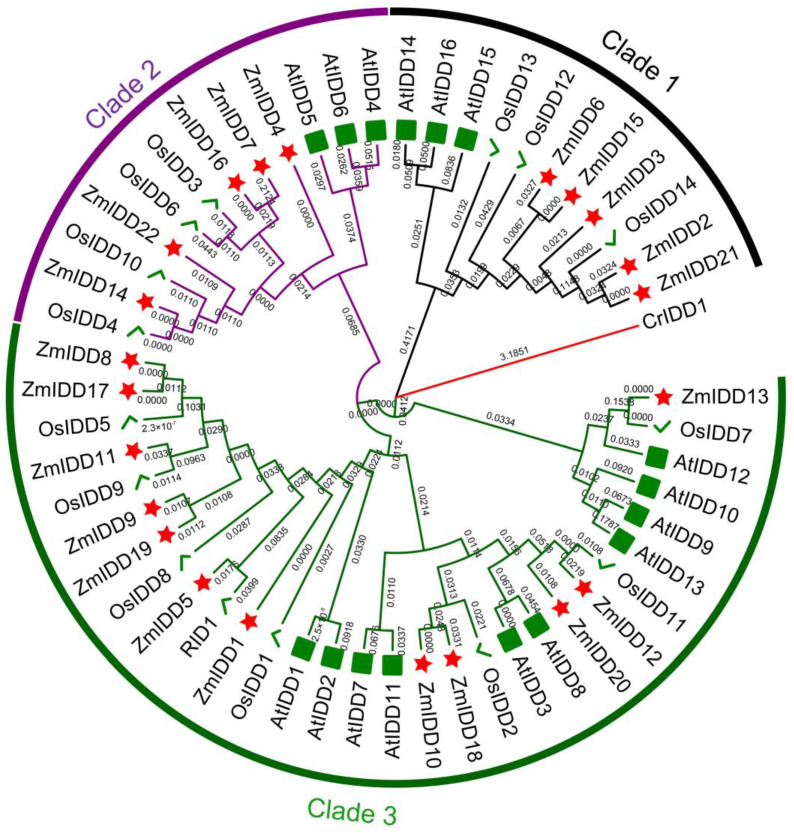
Phylogenetic tree of full-length ZmIDD, AtIDD, and OsIDD proteins using the Maximum Likelihood method based on the JTT matrix-based model. The tree is drawn to scale, with branch lengths measured in the number of substitutions per site. The analysis involved 54 amino acid sequences. All positions containing gaps and missing data were eliminated. There were a total of 94 positions in the final dataset. The different colored arcs indicate subfamilies of the IDD proteins. Different colored shapes represent IDDs from maize (☆), rice (√), and Arabidopsis (☐). IDD in *Chlamydomonas reinhardtii* was selected as an outgroup.

**Figure 2 ijms-24-06185-f002:**
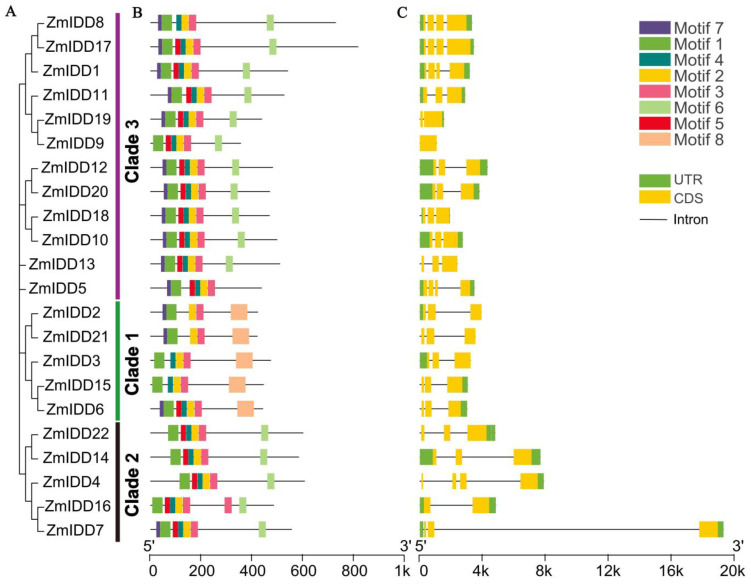
Phylogenetic relationships, architecture of conserved protein motifs, and gene structure in *IDD* genes from maize. (**A**) The phylogenetic tree was constructed based on the full-length sequences of maize IDD proteins using MEGA version 7.0 software. (**B**) The motif compositions of 22 ZmIDD proteins. The motifs were identified using the MEME program. Boxes of different colors represent motifs 1 to 10. The length of the amino acid sequences can be estimated by the scale at the **bottom**. (**C**) Gene structures of 22 *ZmIDD* genes. Yellow boxes represent exons, green boxes represent 5′ or 3′ untranslated regions (UTR), and black lines represent introns. The length of nucleotide sequences of exons/introns/UTRs can be estimated by the scale at the **bottom**.

**Figure 3 ijms-24-06185-f003:**
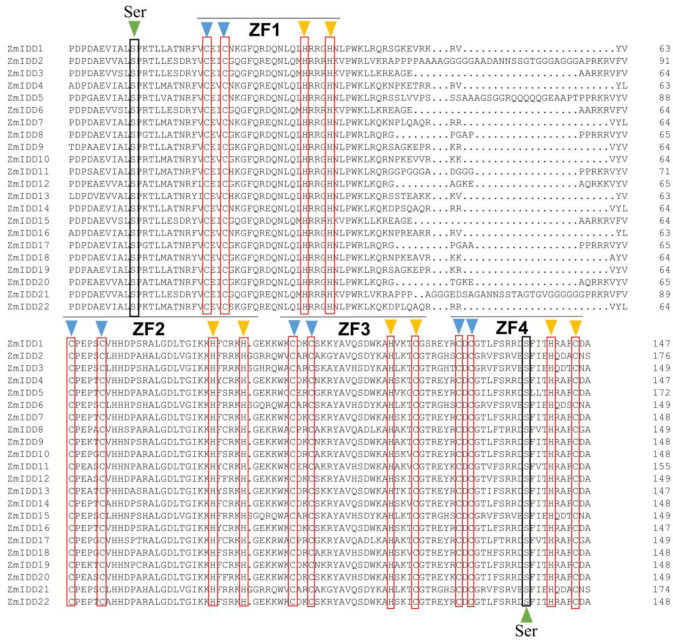
Comparative amino acid sequence alignment of INDETERMINATE DOMAIN (IDD) genes that shows motifs or domains that are conserved in maize. Black boxes mark the positions of cysteines (C, in blue triangles) and histidines (H, in yellow triangles) characterized for each zinc finger.

**Figure 4 ijms-24-06185-f004:**
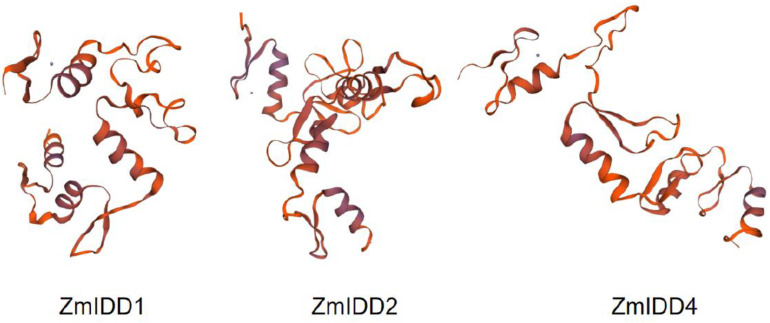
The 3D structure modeling of ZmIDD proteins. The Pymol software was used to create the structural image.

**Figure 5 ijms-24-06185-f005:**
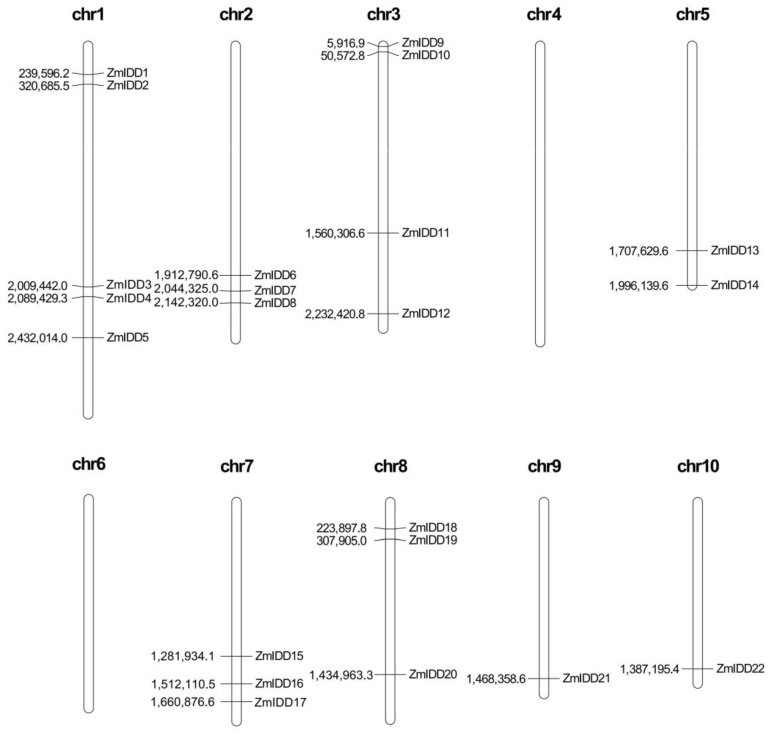
Distribution of *ZmIDD* genes in maize chromosomes. The chromosome numbers are indicated at the **top** of each chromosome image.

**Figure 6 ijms-24-06185-f006:**
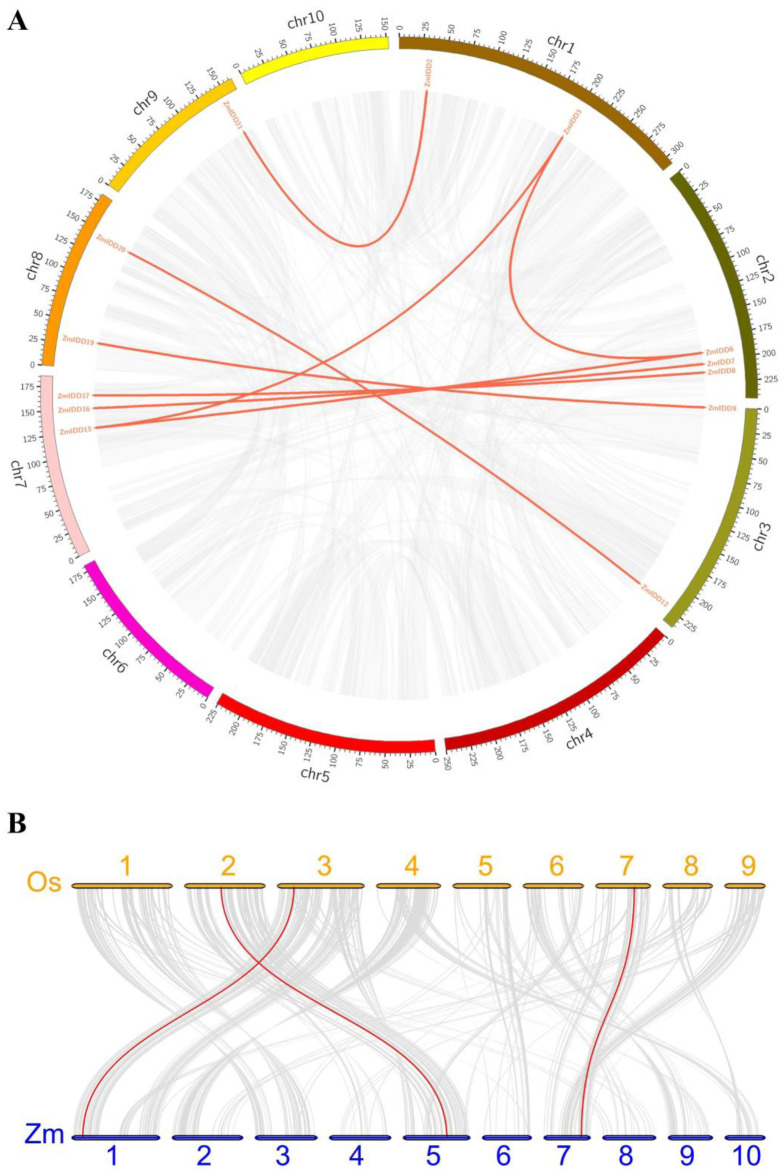
The synteny analysis of *ZmIDD* family genes. (**A**) The synteny analysis of the *ZmIDD* family in maize. Gray lines indicate all synteny blocks in the maize genome. The genes linked by red lines represent homologues. (**B**) Synteny analysis of *IDD* genes between maize and rice. Gray lines: all collinear blocks within maize and other plant genomes. Red lines: the synteny of *IDD* gene pairs. The species names with the prefixes Zm and Os indicate maize and rice, respectively.

**Figure 7 ijms-24-06185-f007:**
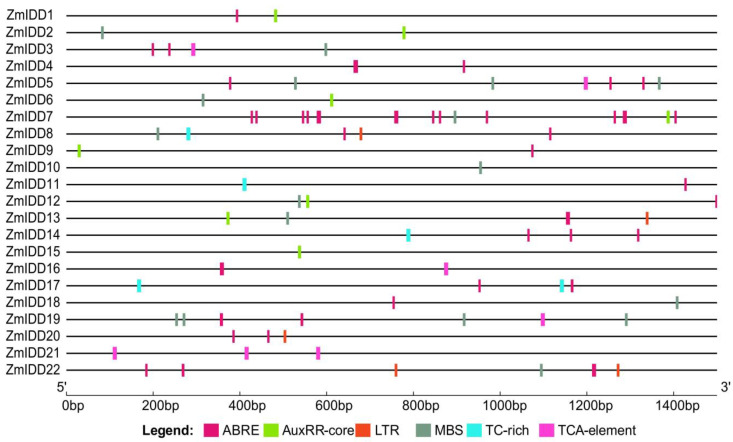
Predicted cis-regulatory elements in *ZmIDD* promoters. Promoter sequences (about 1.5 kb) of 22 *ZmIDD* genes were analyzed by PlantCARE. The upstream length to the translation starting site can be inferred according to the scale at the **bottom**.

**Figure 8 ijms-24-06185-f008:**
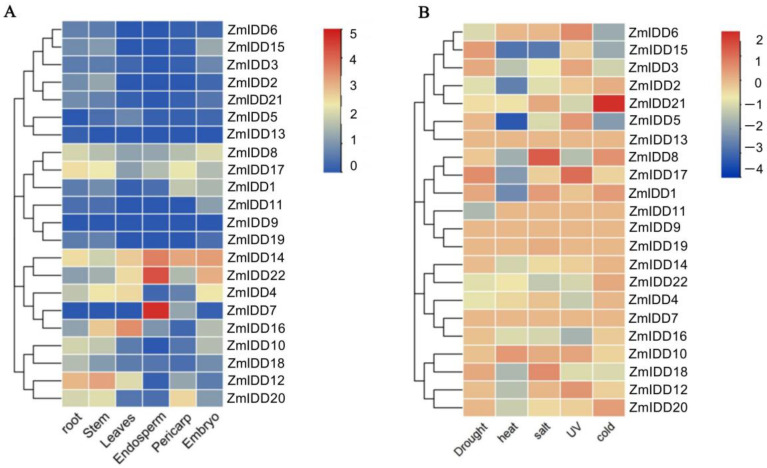
Expression profiles of the *ZmIDD* genes. (**A**) Expression profiles of the *ZmIDD* genes in different tissues. The color scale represents expression data with a row scale. Blue: low expression; red: high expression. (**B**) Expression profiles of the *ZmIDD* genes under different abiotic stresses. Expression data were the ratios to control values. The color scale represents expression levels from upregulation (red) to downregulation (blue).

**Figure 9 ijms-24-06185-f009:**
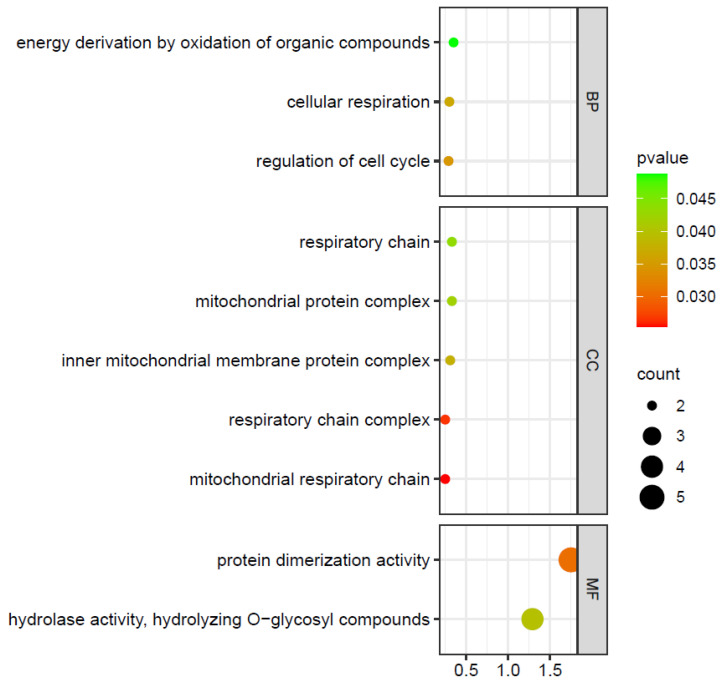
Bubble map of GO enrichment of ZmIDD-target genes. Genes were listed in Appendix A. The *X*-axis represents the Rich Ratio. Rich Ratio = Term Candidate Gene Number/Term Gene Number. The *Y*-axis represents the GO Term. The size of the bubble represents the number of different genes annotated to a GO Term, and the color represents the enriched *p* value. 0 < *p* value < 1. The smaller the *p* value, the more significant the GO enrichment.

**Figure 10 ijms-24-06185-f010:**
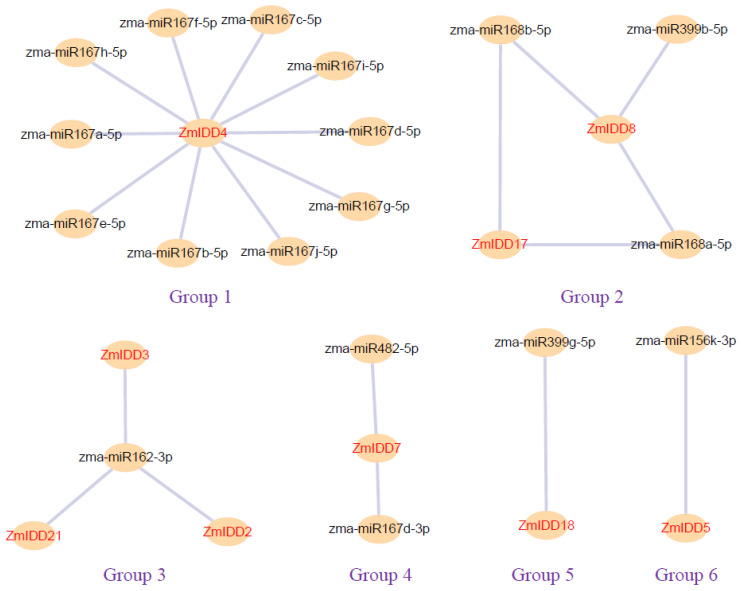
Interaction networks between miRNAs and miRNA-acted *ZmIDDs*. Information on miRNAs and miRNAs-acted *ZmIDDs* is shown in Appendix A.

## Data Availability

All data analyzed during this study are included in this article.

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
