# Peer review of "Comprehensive Analysis of the INDETERMINATE DOMAIN (IDD) Gene Family and Their Response to Abiotic Stress in Zea mays"

_ijms, 2023, doi:10.3390/ijms24076185_

Round 1

Reviewer 1 Report

Review of the article by Xue Feng et al. entitled "Comprehensive Analysis of INDETERMINATE DOMAIN 2 (IDD) Gene Family in Climate Adaption of Zea mays".

The manuscript deals with analysis of specific plant transcription factors playing important roles in plant growth and development, namely the characterization of IDD gene family in maize, a widely cultivated crop throughout the world. The manuscript is generally well written and worthy of publication. However, the article cannot be accepted in its present form and requires revision before final acceptance.

The title does not fully reflect the content of the manuscript. The involvement of IDD genes in response to the influence of external factors on the plant is only one of the cases that are considered in the article.

It is necessary to highlight the Latin plant names in italics.

The phrase "a gene contains one exon" is incorrect (lines 22, 103). Perhaps the authors meant that the gene does not contain an intron. Please clarify and correct.

The designations indicated in the caption to figure 1 do not correspond to those shown in the figure.

I encourage the authors to repeat the phylogenetic analysis using various approaches, not only the Neighbor-Joining method. Also missing from Figure 1 is important information related to branch lengths and node support levels. In addition, it is necessary to describe in more detail the tree reconstruction parameters, in particular, evolutionary models and the outgroup selection.

In sections 2.2 and 2.5, the authors propose to division into several groups that are similarly named (group 1, etc.) but based on different criteria.

In the discussion, it is necessary to clarify how the functionally verified three IDDs (ID1, NKD1, NHD2) correspond to the variants identified in this work.

It is also necessary to reflect in the discussion how the authors solved the issue with paralogs and orthologs, taking into account the different number of whole genome duplications that occurred in the compared plants (maize, rice, and Arabidopsis). This will allow the information obtained by comparing the IDD genes in plants with different degrees of relatedness to be more fully involved in the discussion.

Reviewer 2 Report

This study systematically analyzed IDD gene family in maize, including evolutionary relationship, Cis-element, expression pattern, regulatory networks of ZmIDDs and miRNAs. These results are very meaningful for studying the function of ZmIDDs in the following study. This manuscript is worth be published in IJMS.

There are some flaws in the article that need to be corrected before the article is accepted.

1.      I am not sure if there is a problem with the system, but genes in the article are not in italics.

2.      In Figure 1 and 2, these are labeled “Group1”, “Group2”, ”Group3”, whereas it describe Clade I, II, and III in Line 94, group I, II, III in Line 110. These need to be consistent.

3.      In Figure 1, “Different colour shapes represent IDDs from 98 maize (â–²), rice (√), and Arabidopsis (£)”. The colour shapes doesn't correspond to the Figure.

4.      In Figure 2, black lines represent introns, which should be labeled in Figure 2.

5.      In line 182, “The rest of 6 genes belong to subgroup 2”. “Subgroup” should be labeled in Figure.

6.      There are some tenses and spelling problems that need modify.

Like as,

In Line 135, ZmIDD subfamily â…¡ genes are found on 135 chromosomes 1, 2, 5, 7 and 10; and ZmIDD subfamily â…¢ genes were localized on chromosomes 1, 2, 3, 5, 7 and 8.

In Line 148, we scaned promoter sequence of 22 ZmIDDs to analyze cis-elements.

Round 2

Reviewer 1 Report

The phylogenetic analysis still needs improvement. Please describe in the Methods what evolutionary models and what approaches were used to the tree reconstructions (since three tree reconstruction approaches were used, all of them should be described in the appropriate section of the article). Also it is still missing from Figure 1 important information related to branch lengths. Links to Figure S1 and S2 should be given in the Results.
